

# A direct approach to estimating false discovery rates conditional on covariates

Simina M. Boca[1,2,3] and Jeffrey T. Leek[4]

[1] Innovation Center for Biomedical Informatics, Georgetown University Medical Center, Washington, D.C., USA
[2] Department of Oncology, Georgetown University Medical Center, Washington, D.C., USA
[3] Department of Biostatistics, Bioinformatics & Biomathematics, Georgetown University Medical Center, Washington, D.C., USA
[4] Department of Biostatistics, Johns Hopkins Bloomberg School of Public Health, Baltimore, MD, USA

## ABSTRACT

Modern scientific studies from many diverse areas of research abound with multiple hypothesis testing concerns. The false discovery rate (FDR) is one of the most commonly used approaches for measuring and controlling error rates when performing multiple tests. Adaptive FDRs rely on an estimate of the proportion of null hypotheses among all the hypotheses being tested. This proportion is typically estimated once for each collection of hypotheses. Here, we propose a regression framework to estimate the proportion of null hypotheses conditional on observed covariates. This may then be used as a multiplication factor with the Benjamini–Hochberg adjusted $p$-values, leading to a plug-in FDR estimator. We apply our method to a genome-wise association meta-analysis for body mass index. In our framework, we are able to use the sample sizes for the individual genomic loci and the minor allele frequencies as covariates. We further evaluate our approach via a number of simulation scenarios. We provide an implementation of this novel method for estimating the proportion of null hypotheses in a regression framework as part of the Bioconductor package swfdr.

## INTRODUCTION

Multiple testing is a ubiquitous issue in modern scientific studies. Microarrays (*Schena et al., 1995*), next-generation sequencing (*Shendure & Ji, 2008*), and high-throughput metabolomics (*Lindon, Nicholson & Holmes, 2011*) make it possible to simultaneously test the relationship between hundreds or thousands of biomarkers and an exposure or outcome of interest. These problems have a common structure consisting of a collection of variables, or features, for which measurements are obtained on multiple samples, with a hypothesis test being performed for each feature.

When performing thousands of hypothesis tests, one of the most widely used frameworks for controlling for multiple testing is the false discovery rate (FDR). For a fixed unknown parameter $\mu$, and testing a single null hypothesis $H_0: \mu = \mu_0$ vs some alternative hypothesis, for example, $H_1: \mu = \mu_1$, the null hypothesis may either truly hold or not

Corresponding authors
Simina M. Boca,
smb310@georgetown.edu
Jeffrey T. Leek, jtleek@gmail.com

**Table 1 Outcomes of testing multiple hypotheses.**

|  | Fail to reject null | Reject null | Total |
|---|---|---|---|
| Null true | $U$ | $V$ | $m_0$ |
| Null false | $T$ | $S$ | $m-m_0$ |
|  | $m-R$ | $R$ | $m$ |

for each feature. Additionally, the test may lead to $H_0$ either being rejected or not being rejected. Thus, when performing $m$ hypothesis tests for $m$ different unknown parameters, Table 1 shows the total number of outcomes of each type, using the notation from the work of *Benjamini & Hochberg (1995)*. We note that $U$, $T$, $V$, and $S$, and as a result, also $R = V + S$, are random variables, while $m_0$, the number of null hypotheses, is fixed and unknown.

The FDR, introduced by *Benjamini & Hochberg (1995)*, is the expected fraction of false discoveries among all discoveries. The FDR depends on the overall fraction of null hypotheses, namely $\pi_0 = \frac{m_0}{m}$. This proportion can also be interpreted as the a priori probability that a null hypothesis is true, $\pi_0$.

When estimating the FDR, incorporating an estimate of $\pi_0$ can result in a more powerful procedure compared to the original procedure of *Benjamini & Hochberg (1995)* (BH); moreover, as $m$ increases, the estimate of $\pi_0$ improves, which means that the power of the multiple-testing approach does not necessarily decrease when more hypotheses are considered (*Storey, 2002*). The popularity of this approach can be seen in the extensive use of the `qvalue` package (*Storey et al., 2015*), which implements this method, which is among the top 5% most downloaded Bioconductor packages, having been downloaded more than 78,000 times in 2017.

Most modern adaptive FDR procedures rely on an estimate of $\pi_0$ using the data from all tests being performed. But additional information, in the form of meta-data, may be available to aid the decision about whether to reject the null hypothesis for a particular feature. The concept of using these feature-level covariates, which may also be considered "prior information," arose in the context of $p$-value weighting (*Genovese, Roeder & Wasserman, 2006*). We focus on an example from a genome-wide association study (GWAS) meta-analysis, in which millions of genetic loci are tested for associations with an outcome of interest—in our case body mass index (BMI) (*Locke et al., 2015*). Different loci may not all be genotyped in the same individuals, leading to loci-specific sample sizes.

Additionally, each locus will have a different population-level frequency. Thus, the sample sizes and the frequencies may be considered as covariates of interest. Other examples exist in set-level inference, including gene set analysis, where each set has a different fraction of false discoveries. Adjusting for covariates independent of the data conditional on the truth of the null hypothesis has also been shown to improve power in RNA-seq, eQTL, and proteomics studies (*Ignatiadis et al., 2016*).

In this paper, we develop and implement an approach for estimating FDRs conditional on covariates and apply it to a genome-wide analysis study. Specifically, we seek to better understand the impact of sample sizes and allele frequencies in the BMI GWAS data analysis by building on the approaches of *Benjamini & Hochberg (1995)*, *Efron et al. (2001)*,

and *Storey (2002)* and the more recent work of *Scott et al. (2015)*, which frames the concept of FDR regression and extends the concepts of FDR and $\pi_0$ to incorporate covariates, represented by additional meta-data. Our focus will be on estimating the covariate-specific $\pi_0$, which will then be used in a plug-in estimator for the FDR, similar to the work of *Storey (2002)*. We thus provide a more direct approach to estimating the FDR conditional on covariates and compare our estimates to those of *Scott et al. (2015)*, as well as to the BH and *Storey (2002)* approaches. Our method for estimating the covariate-specific $\pi_0$ is implemented in the Bioconductor package `swfdr` (https://bioconductor.org/packages/release/bioc/html/swfdr.html). Similar very recent approaches include work by *Li & Barber (2017)* and *Lei & Fithian (2018)*, which also estimate $\pi_0$ based on existing covariates, using different approaches. The approach of *Ignatiadis et al. (2016)* considers *p*-value weighting but conservatively estimates $\pi_0 \equiv 1$. An overview of the differences between these various approaches for incorporating meta-data and the relationships between them is provided by *Ignatiadis & Huber (2018)*.

The remainder of the paper is organized as follows: We first introduce the motivating case study, a BMI GWAS meta-analysis, which will be discussed throughout the paper. We then review the definitions of FDR and $\pi_0$ and their extensions to consider conditioning on specific covariates; discuss estimation and inference procedures in our FDR regression framework, provide a complete algorithm, and apply it to the GWAS case study; and describe results from a variety of simulation scenarios. Finally, we provide our statement of reproducibility, followed by the discussion. Special cases, theoretical properties of the estimator, and proofs of the results can be found in the Supplementary Materials.

## MOTIVATING CASE STUDY: ADJUSTING FOR SAMPLE SIZE AND ALLELE FREQUENCY IN GWAS META-ANALYSIS

As we have described, there are a variety of situations where meta-data could be valuable for improving the decision of whether a hypothesis should be rejected in a multiple testing framework, our focus being on an example from the meta-analysis of data from a GWAS for BMI (*Locke et al., 2015*). Using standard approaches such as that of *Storey (2002)*, we can estimate the fraction of single nucleotide polymorphisms (SNPs)—genomic positions (loci) which show between-individual variability—which are not truly associated with BMI and use it in an adaptive FDR procedure. However, our proposed method allows further modeling of this fraction as a function of additional study-level meta-data.

In a GWAS, data are collected for a large number of SNPs in order to assess their associations with an outcome or trait of interest (*Hirschhorn & Daly, 2005*). Each person usually has one copy of the DNA at each SNP inherited from their mother and one from their father. At each locus there are usually one of two types of DNA, called alleles, that can be inherited, which we denote *A* and *a*. In general, *A* refers to the variant that is more common in the population being studied and *a* to the variant that is less common, usually called the minor allele. Each person has a genotype for that SNP of the

form *AA*, *Aa*, or *aa*. For example, for a particular SNP, of the four possible DNA nucleotides, adenine, guanine, cytosine, and thymine, an individual may have either a cytosine (C) or a thymine (T) at a particular locus, leading to the possible genotypes CC, CT, and TT. If the C allele is less common in the population, then C is the minor allele. The number of copies of *a*, which is between 0 and 2, is often assumed to follow a binomial distribution, which generally differs between SNPs.

Typically, a GWAS involves performing an association test between each SNP and the outcome of interest by using a regression model, including the calculation of a *p*-value. While GWAS studies are often very large, having sample sizes of tens of thousands of individuals genotyped at hundreds of thousands of SNPs, due to the small effect sizes being detected, meta-analyses combining multiple studies are often considered (*Neale et al., 2010*; *Hirschhorn & Daly, 2005*). In these studies, the sample size may not be the same for each SNP, for example, if different individuals are measured with different technologies which measure different SNPs. Sample size is thus a covariate of interest, as is the minor allele frequency (MAF) of the population being studied, which will also vary between SNPs. The power to detect associations increases with MAF. This is related to the idea that logistic regression is more powerful for outcomes that occur with a frequency close to 0.5. Our approach will allow us to better quantify this dependence in order to guide the planning of future studies and improve understanding of already-collected data.

We consider data from the Genetic Investigation of ANthropometric Traits (GIANT) consortium, specifically the GWAS for BMI (*Locke et al., 2015*). The GIANT consortium performed a meta-analysis of 339,224 individuals measuring 2,555,510 SNPs and tested each for association with BMI. A total of 322,154 of the individuals considered by *Locke et al. (2015)* are of European descent and the study uses the HapMap CEU population—which consists of individuals from Utah of Northern and Western European ancestry (*International HapMap Consortium, 2007*)—as a reference. We used the set of results from the GIANT portal at http://portals.broadinstitute.org/collaboration/giant/index.php/GIANT_consortium_data_files, which provides the SNP names and alleles, effect allele frequencies (EAFs) in the HapMap CEU population and results from the regression-based association analyses for BMI, presented as beta coefficients, standard errors, *p*-values, and sample size for each SNP.

We removed the SNPs that had missing EAFs, leading to 2,500,573 SNPs. For these SNPs, the minimum sample size considered was 50,002, the maximum sample size 339,224, and the median sample size 235,717—a relatively wide range. Figure 1 shows the dependence of *p*-values on sample sizes within this dataset. As we considered the MAF to be a more intuitive covariate than the EAF, we also converted EAF values >0.5 to MAF $= 1 - $ EAF and changed the sign of the beta coefficients for those SNPs. The MAFs spanned the entire possible range from 0 to 0.5, with a median value of 0.208.

## COVARIATE-SPECIFIC $\pi_0$ AND FDR

We will now review the main concepts behind the FDR and the a priori probability that a null hypothesis is true, and consider the extension to the covariate-specific FDR

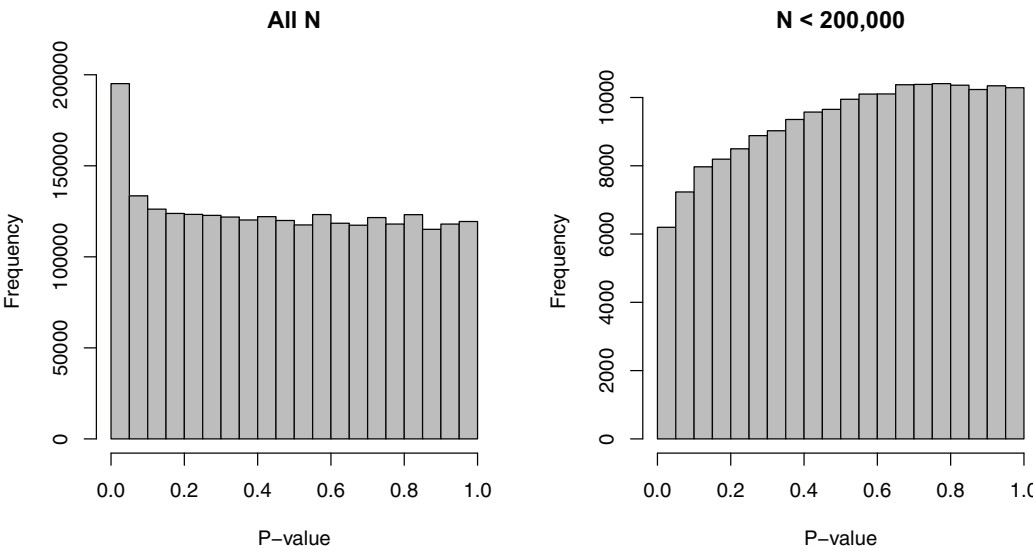

**Figure 1 Histograms of *p*-values for the SNP-BMI tests of association from the GIANT consortium.**
(A) shows the distribution for all sample sizes $N$ (2,500,573 SNPs), while (B) shows the subset $N < 200,000$
(187,114 SNPs).

and the covariate-specific a priori probability. A natural mathematical definition of the
FDR would be:

$$\text{FDR} = E\left(\frac{V}{R}\right).$$

However, $R$ is a random variable that can be equal to 0, so the version that is generally
used is:

$$\text{FDR} = E\left(\frac{V}{R}\bigg| R > 0\right)\Pr(R > 0), \tag{1}$$

namely the expected fraction of false discoveries among all discoveries, conditional
on at least one rejection, multiplied by the probability of making at least one
rejection.

We index the $m$ null hypotheses being considered by $1 \le i \le m$: $H_{01}, H_{02}, \ldots, H_{0m}$. For
each $i$, the corresponding null hypothesis $H_{0i}$ can be considered as being about a binary
parameter $\theta_i$, such that:

$$\theta_i = 1\ (H_{0i}\ \text{false}).$$

Thus, assuming that $\theta_i$ are identically distributed, the a priori probability that a feature
is null is:

$$\pi_0 = \Pr(\theta_i = 0). \tag{2}$$

For the GWAS meta-analysis dataset, $\pi_0$ represents the proportion of SNPs which are
not truly associated with BMI or, equivalently, the prior probability that any of the SNPs is
not associated with BMI.

---

**Algorithm 1** Estimation and inference for $\hat{\pi}_0(\mathbf{x}_i)$ and $\widehat{FDR}(x_i)$

(a) Obtain the *p*-values $P_1, P_2, \ldots, P_m$, for the *m* hypothesis tests.

(b) For a given threshold $\lambda$, obtain $Y_i = 1(P_i > \lambda)$ for $1 \leq i \leq m$.

(c) Estimate $E(Y_i|\mathbf{X}_i = \mathbf{x}_i)$ via logistic regression using a design matrix $\mathbf{Z}$ and $\pi_0(\mathbf{x}_i)$ by:

$$\hat{\pi}_0^\lambda(\mathbf{x}_i) = \frac{\hat{E}(Y_i|\mathbf{X}_i = \mathbf{x}_i)}{1 - \lambda}, \tag{3}$$

thresholded at 1 if necessary.

(d) Smooth $\hat{\pi}_0^\lambda(\mathbf{x}_i)$ over a series of thresholds $\lambda \in (0, 1)$ to obtain $\hat{\pi}_0(\mathbf{x}_i)$, by taking the smoothed value at the largest threshold considered. Take the minimum between each value and 1 and the maximum between each value and 0.

(e) Take *B* bootstrap samples of $P_1, P_2, \ldots, P_m$ and calculate the bootstrap estimates $\hat{\pi}_0^b(\mathbf{x}_i)$ for $1 \leq b \leq B$ using the procedure described above.

(f) Form a $1 - \alpha$ confidence interval for $\hat{\pi}_0(\mathbf{x}_i)$ by taking the $1 - \alpha/2$ quantile of the $\hat{\pi}_0^b(\mathbf{x}_i)$ as the upper confidence bound, the lower confidence bound being $\alpha/2$.

(g) Obtain an $\widehat{FDR}(\mathbf{x}_i)$ by multiplying the BH adjusted *p*-values by $\hat{\pi}_0(\mathbf{x}_i)$.

---

We now extend $\pi_0$ and FDR to consider conditioning on a set of covariates concatenated in a column vector $\mathbf{X}_i$ of length $c$, possibly with $c = 1$:

$$\pi_0(\mathbf{x}_i) = \Pr(\theta_i = 0|\mathbf{X}_i = \mathbf{x}_i),$$

$$\text{FDR}(\mathbf{x}_i) = E\left(\frac{V}{R}\bigg|R > 0, \mathbf{X}_i = \mathbf{x}_i\right)\Pr(R > 0|\mathbf{X}_i = \mathbf{x}_i).$$

## ALGORITHM FOR PERFORMING ESTIMATION AND INFERENCE FOR COVARIATE-SPECIFIC $\pi_0$ AND FDR

Assuming that a hypothesis test is performed for each feature $i$, summarized by a *p*-value $P_i$, Algorithm 1 can be used to obtain estimates of $\pi_0(\mathbf{x}_i)$ and FDR$(\mathbf{x}_i)$, denoted by $\hat{\pi}_0(\mathbf{x}_i)$ and $\widehat{FDR}(\mathbf{x}_i)$, and perform inference. In Step (c) $\mathbf{Z}$ is a $m \times p$ design matrix with $p < m$ and rank$(\mathbf{Z}) = d \leq p$, which can either be equal to $\mathbf{X}$—the matrix of dimension $m \times (c + 1)$, which has the $i$th row consisting of $(1 \; \mathbf{X}_i^T)$—or include additional columns that are functions of the covariates in $\mathbf{X}$, such as polynomial or spline terms. The estimator is similar to:

$$\hat{\pi}_0 = \frac{\frac{\sum_{i=1}^m Y_i}{m}}{1 - \lambda} = \frac{m - R}{(1 - \lambda)m}, \tag{4}$$

which is used by *Storey (2002)* for the case without covariates. In Step (c) we focus on maximum likelihood estimation of $E(Y_i|\mathbf{X}_i = \mathbf{x}_i)$, assuming a logistic model. A linear regression approach would be a more direct generalization of Storey's method, but a logistic model is more natural for estimating means between 0 and 1. In particular, we note that a linear regression approach would amplify relatively small differences between large values of $\pi_0(\mathbf{x}_i)$, which are likely to be common in many scientific situations, especially when considering GWAS, where one may expect a relatively low number of SNPs to be

truly associated with the outcome of interest. In the `swfdr` package, we provide users the choice to estimate $\pi_0(\mathbf{x}_i)$ via either the logistic or linear regression model. In Step (d), we consider smoothing over a series of thresholds to obtain the final estimate, as done by *Storey & Tibshirani (2003)*. In particular, in the remainder of this manuscript, we used cubic smoothing splines with three degrees of freedom over the series of thresholds 0.05, 0.10, 0.15, ..., 0.95, following the example of the `qvalue` package (*Storey et al., 2015*), with the final estimate being the smoothed value at $\lambda = 0.95$. We note that the final Step (g) results in a simple plug-in estimator for FDR$(\mathbf{x}_i)$.

We provide further details in the Supplementary Materials: In Section S1, we present the assumptions and main results used to derive Algorithm 1; in Section S2, we detail how the case of no covariates and the case where the features are partitioned into sets, such as in the work of *Hu, Zhao & Zhou (2010)*, can be seen as special cases of our more general framework when the linear regression approach is applied; in Section S3 we provide theoretical results for this estimator; in Section S4, we present proofs of the analytical results. We note that a major assumption is that *conditional on the null, the p-values do not depend on the covariates*. Our theoretical results are based on the more restrictive assumption that the null *p*-values have a Uniform(0,1) distribution, whereas the distribution of the alternative *p*-values may depend on the covariates. This means that the probability of a feature being from one of the two distributions depends on the covariates but the actual test statistic and *p*-value under the null do not depend on the covariates further.

The model we considered for the GWAS meta-analysis dataset models the SNP-specific sample size using natural cubic splines, in order to allow for sufficient flexibility. It also considers three discrete categories for the CEU MAFs, corresponding to cuts at the 1/3 and 2/3 quantiles, leading to the intervals (0.000, 0.127) (838,070 SNPs), (0.127, 0.302) (850,600 SNPs), and (0.302, 0.500) (811,903 SNPs).

Figure 2 shows the estimates of $\pi_0(\mathbf{x}_i)$ plotted against the SNP-specific sample size $N$ for the data analysis, stratified by the CEU MAFs for a random subset of 50,000 SNPs. We note that the results are similar for $\lambda = 0.8$, $\lambda = 0.9$, and for the final smoothed estimate. A 95% bootstrap confidence interval based on 100 iterations is also shown for the final smoothed estimate. Our approach is compared to that of *Scott et al. (2015)*, which assumes that the test statistics are normally distributed. We considered both the theoretical and empirical null Empirical Bayes (EB) estimates from *Scott et al. (2015)*, implemented in the `FDRreg` package (*Scott, Kass & Windle, 2015*). The former assumes a $N(0,1)$ distribution under the null, while the latter estimates the parameters of the null distribution. Both approaches show similar qualitative trends to our estimates, although the empirical null tends to result in much higher values over the entire range of $N$, while the theoretical null leads to lower values for smaller $N$ and larger or comparable values for larger $N$. Our results are consistent with intuition—larger sample sizes and larger MAFs lead to a smaller fraction of SNPs estimated to be null. They do, however, allow for improved quantification of this relationship: For example, we see that the range for $\hat{\pi}_0(\mathbf{x}_i)$ is relatively wide ((0.697, 1) for the final smoothed estimate), while the smoothed estimate of $\pi_0$ without covariates—obtained via the *Storey (2002)*

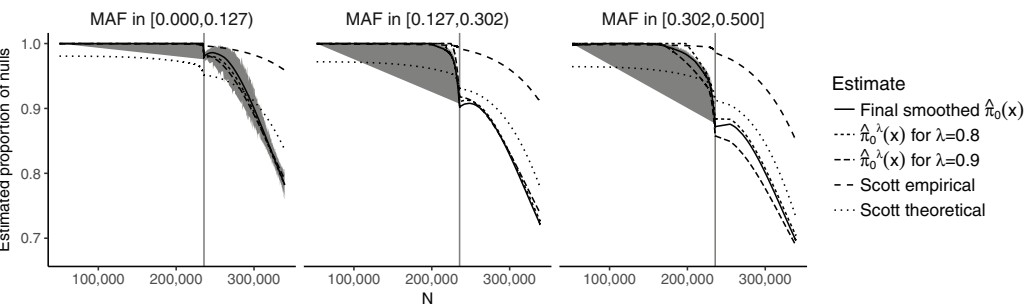

**Figure 2** **Plot of the estimates of $\pi_0(x_i)$ against the sample size $N$, stratified by the MAF categories for a random subset of 50,000 SNPs.** The 90% bootstrap intervals for the final smoothed estimates using our approach—based on 100 iterations—are shown in gray. The vertical line represents the median sample size.

approach—is 0.949. In the `swfdr` package, we include a subset of the data—for 50,000 randomly selected SNPs—and show how to generate plots similar to Fig. 2. Users may of course consider the full dataset and reproduce our entire analysis (see Section 6 on reproducibility below.)

The results for the number of SNPs with estimated FDR ≤0.05 are given in Table S1. Our approach results in a slightly larger number of discoveries compared to the *Storey (2002)* and *Benjamini & Hochberg (1995)* approaches. Due to the plug-in approaches of both our procedure and that of *Storey (2002)*, all the discoveries obtained using the *Benjamini & Hochberg (1995)* method are also present in our approach. The total number of shared discoveries between our method and that of *Storey (2002)* is 12,740. The *Scott et al. (2015)* approaches result in either a substantially larger number of discoveries (theoretical null) or a substantially smaller number of discoveries (empirical null). In particular, the number of discoveries for the empirical null is also much smaller than when using the *Benjamini & Hochberg (1995)* approach. The overlap between the theoretical null and the *Benjamini & Hochberg (1995)* method is 12,251; between the theoretical null and our approach it is 13,119.

## SIMULATIONS

We consider simulations to evaluate how well $\hat{\pi}_0(\mathbf{x}_i)$ estimates $\pi_0(\mathbf{x}_i)$, as well as the usefulness of our plug-in estimator, $\widehat{\text{FDR}}(\mathbf{x}_i)$, in terms of both controlling the true FDR and having good power—measured by the true positive rate (TPR)—under a variety of scenarios. We consider a nominal FDR value of 5%, meaning that any test with an FDR less than or equal to 5% is considered a discovery. In each simulation, the FDR is calculated as the fraction of truly null discoveries out of the total number of discoveries and the TPR is the fraction of truly alternative discoveries out of the total number of truly alternative features. In the case of no discoveries, the FDR is estimated to be 0.

We focus on five different possible functions $\pi_0(\mathbf{x}_i)$, shown in Fig. 3. Scenario I considers a flat function $\pi_0 = 0.9$, to illustrate a case where there is no dependence on covariates and scenarios II–IV are similar to the BMI GWAS meta-analysis. Scenarios II–IV are chosen to be similar to the BMI GWAS meta-analysis. Thus, scenario II is a smooth function of one variable similar to Fig. 2 (MAF in [0.302, 0.500]), scenario III is a

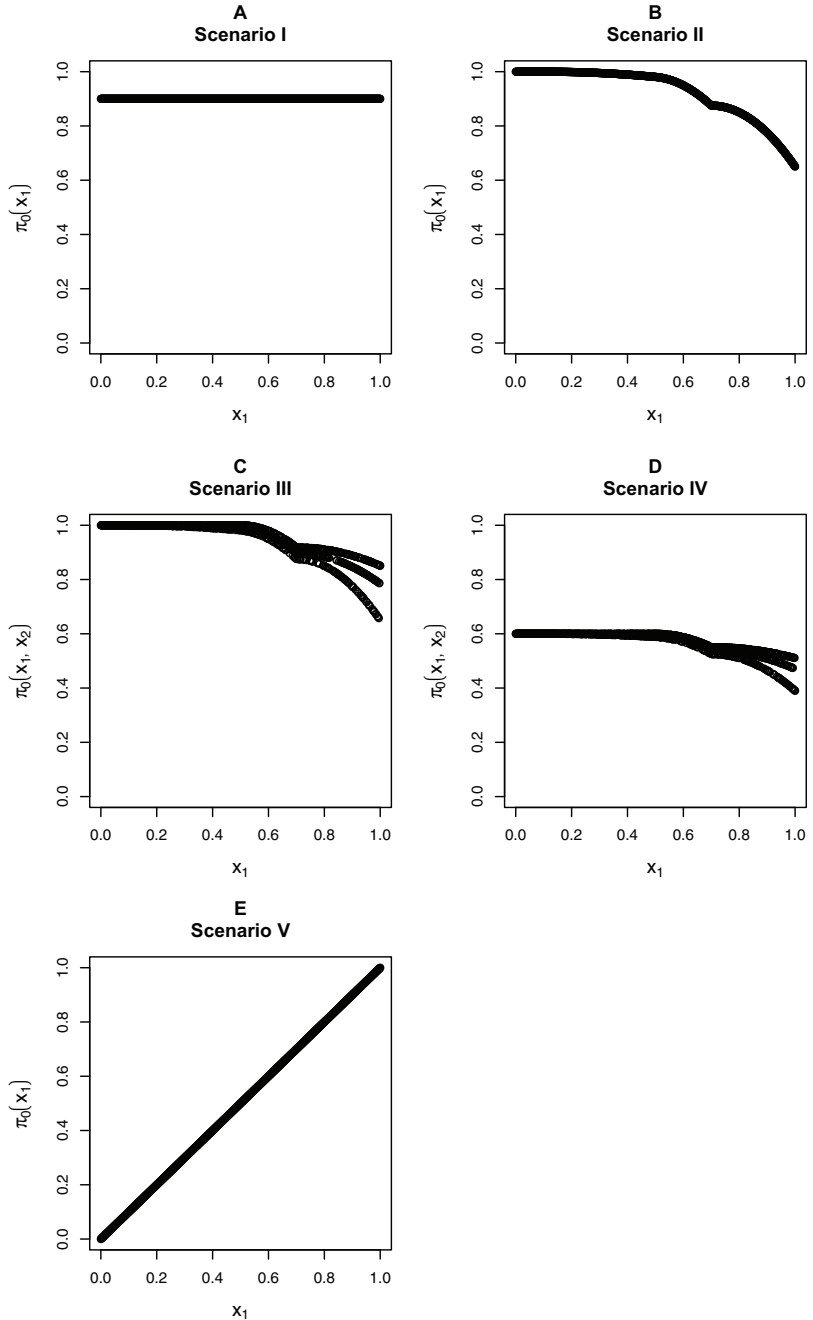

**Figure 3 The five simulation scenarios considered for $\pi_0(x_i)$.** (A) Scenario I. (B) Scenario II. (C) Scenario III. (D) Scenario IV. (E) Scenario V. Scenarios I, II, and V consider smooth functions of a single covariate, whereas scenarios III and IV consider smooth functions of a single covariate ($x_1$) within categories of a second covariate ($x_2$).

function which is smooth in one variable within categories of a second variable—similar to the stratification of SNPs within MAFs—and scenario IV is the same function as in scenario III multiplied by 0.6, to show the effect of having much lower fractions of null hypotheses, respectively, higher fractions of alternative hypotheses. Finally, scenario V is chosen to represent a case where the covariate is very informative; specifically, it

represents the linear function $\pi_0(x_1) = x_1$. The exact functions are given in Section S5 of the Supplementary Materials for this paper. For scenarios I and V we focus on fitting a model that is linear in $x_1$ on the logistic scale, whereas for scenarios II–IV we consider a model that is linear in $x_1$ and a model that fits cubic splines with three degrees of freedom for $x_1$, both on the logistic scale. For scenarios III and IV, all models also consider different coefficients for the categories of $x_2$.

Our first set of simulations considers independent test statistics with either $m = 1,000$ or $m = 10,000$ features. For each simulation run, we first randomly generated whether each feature was from the null or alternative distribution, so that the null hypothesis was true for the features for which a success was drawn from the Bernoulli distribution with probability $\pi_0(\mathbf{x}_i)$. Within each scenario, we allowed for different distributions for the alternative test statistics/$p$-values: beta distribution for the $p$-values or normal, $t$, or Chi-squared distribution for the test statistics. For the beta distribution, we generated the alternative $p$-values directly from a Beta(1,20) distribution and the null $p$-values from a Unif(0,1) distributions. For the other simulations, we first generated the test statistics, then calculated the $p$-values from them. For the normally distributed and $t$-distributed test statistics, we drew the means $\mu_i$ of approximately half the alternative features from a $N(\mu = 3, \sigma^2 = 1)$, with the remaining alternative features from a $N(\mu = -3, \sigma^2 = 1)$ distribution, with the mean of the null features being 0. We then drew the actual test statistic for feature $i$ from either a $N(\mu = \mu_i, \sigma^2 = 1)$ or $T(\mu = \mu_i, df = 10)$ distribution ($df$ = degrees of freedom). Note that 10 degrees of freedom for a $t$-distribution is obtained from a two-sample $t$-test with six samples per group, assuming equal variances in the groups. We also considered Chi-squared test statistics with either one degree of freedom (corresponding to a test of independence for a $2 \times 2$ table) or four degrees of freedom (corresponding to a test of independence for a $3 \times 3$ table). In this case, we first drew the non-centrality parameter ($ncp_i$) from the square of a $N(\mu = 3, \sigma^2 = 1)$ distribution for the alternative and took it to be 0 for the null, then generated the test statistics from $\chi^2(ncp_i = \mu_i, df = 1 \text{ or } 4)$.

Figure 4 considers the case of normally-distributed test statistics with $m = 1,000$ features. Each panel displays the true function $\pi_0(x_i)$ along with the empirical means of $\hat{\pi}_0(\mathbf{x}_i)$, estimated from our approach (BL = Boca–Leek), the *Storey (2002)* approach as implemented in the `qvalue` package (*Storey et al., 2015*), and the theoretical approach in *Scott et al. (2015)* (Scott T), implemented in the `FDRreg` package. For both our approach and the Scott T approach, we plotted both the results for both the linear the cubic spline models. For scenario I ($\pi_0 = 0.9$), the results for the three methods are nearly indistinguishable. For scenarios II–V, the covariates are informative, with both of our approach and the Scott T approach being able to flexibly model the dependence of the function $\pi_0$ on $\mathbf{x}_i$. For scenarios II–III, our approach does show some amount of anti-conservative behavior for the higher values of $\pi_0$, especially for the spline model fit. For scenario V, both our approach and the Scott T approach show a clear increase of $\pi_0$ with $x_{i1}$; given that we are using a logistic model, we are not expecting an exact linear estimate. Figure S1 presents the $m = 1,000$ case with $t$-distributed test statistics. The *Scott et al. (2015)* methods use $z$-values, as opposed to the other methods, which

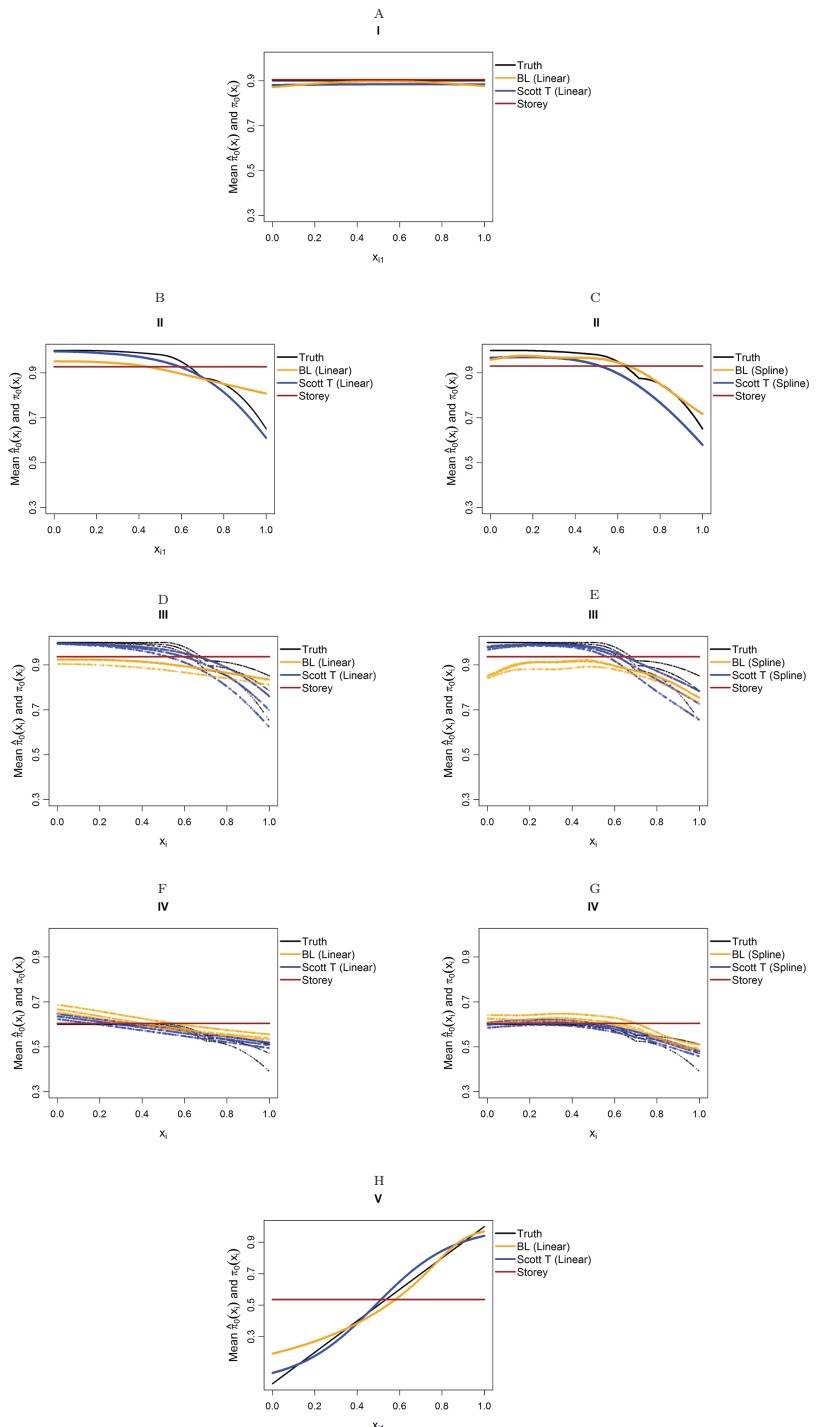

**Figure 4 Simulation results for $m = 1,000$ features and normally-distributed independent test statistics.** Plots show the true function $\pi_0(\mathbf{x}_i)$ in black and the empirical means of $\hat{\pi}_0(\mathbf{x}_i)$, assuming different modeling approaches in orange (for our approach, Boca–Leek = BL), blue (for the Scott approach with the theoretical null = Scott T), and brown (for the Storey approach). The scenarios considered are those in Fig. 3. (A–H) each consider a different combination of scenario (marked I–V) and estimation approach (linear or spline terms for BL and Scott in B–G, linear terms only in A and H).

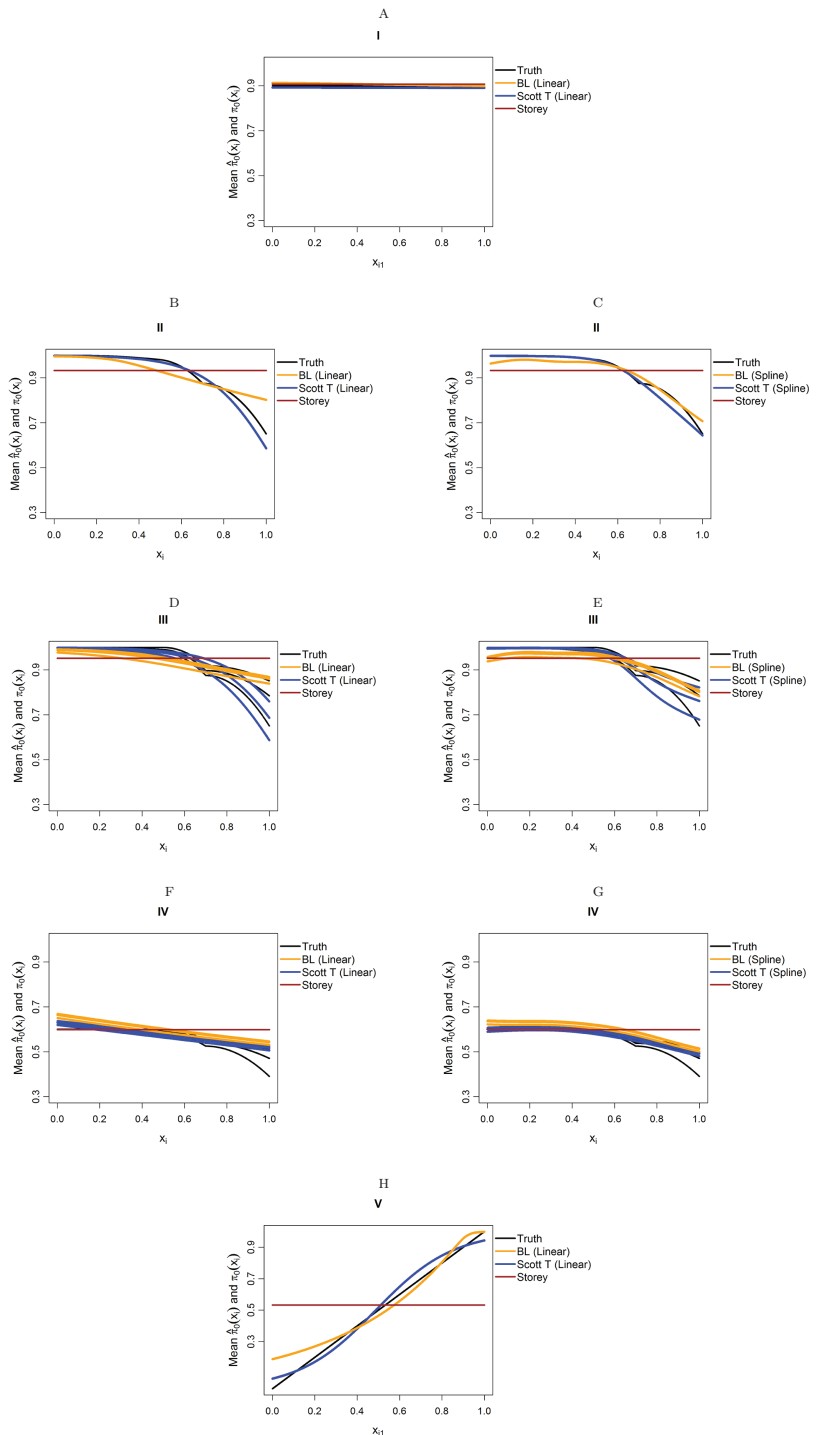

**Figure 5 Simulation results for $m = 10,000$ features and normally-distributed independent test statistics.** Plots show the true function $\pi_0(x_i)$ in black and the empirical means of $\hat{\pi}_0(x_i)$, assuming different modeling approaches in orange (for our approach, Boca–Leek = BL), blue (for the Scott approach with the theoretical null = Scott T), and brown (for the Storey approach.) The scenarios considered are those in Fig. 3. (A–H) each consider a different combination of scenario (marked I–V) and estimation approach (linear or spline terms for BL and Scott in B–G, linear terms only in A and H).

use $p$-values; as a result, in this case we input the $t$-statistics into the Scott T approach, leading to a more pronounced anti-conservative behavior in some cases. This is not the case for our approach or the Storey approach, which rely on $p$-values. Figure 5 and Figure S2 are similar to Fig. 4 and Fig. S1, but consider the $m = 10,000$ case instead. We note that we see less anti-conservativeness for $m = 10,000$, as the estimation is based on a higher number of features. For all these simulation frameworks, we note that for scenario I, the overall mean across all simulations for our method was between 0.88 and 0.91, very close to the true value of 0.9.

Tables 2 and 3 show the results for the FDR and TPR of the plug-in estimators for the scenarios from Fig. 3. In addition to our method, Scott T, Storey, and BH, we consider the null EB approach of *Scott et al. (2015)* (Scott E). We only report the results for the Scott T and Scott E approaches for the cases of the $z$-statistics and $t$-statistics, where these are inputted directly in the methods implemented in the `FDRreg` package. We see in Tables 2 and 3 that our approach had a true FDR close to the nominal value of 5% in most scenarios. As expected, its performance is better for $m = 10,000$, with some slight anticonservative behavior for $m = 1,000$, especially when considering the spline models, also noted from the plots of $\hat{\pi}_0(\mathbf{x}_i)$. We also include the results when fitting splines for our method and the Scott approaches for scenarios I and V and $m = 1,000$ in Table S2.

The *Scott et al. (2015)* approaches perform the best in the case where the test statistics are normally distributed, as expected. In particular, the FDR control of the theoretical null approach is also close to the nominal level and the TPR can be 15% higher in absolute terms than that of our approach for scenarios II and III. The empirical null performs less well. However, the *Scott et al. (2015)* approaches lose control of the FDR when used with $t$-statistics and are not applicable to the other scenarios. We always see a gain in power for our method over the BH approach, however, it is often marginal (1–3%) for scenarios I–III, which have relatively high values of $\pi_0(\mathbf{x}_i)$, which is to be expected, since BH in essence assumes $\pi_0(\mathbf{x}_i) \equiv 1$. For scenario IV, however, the average TPR may increase by as much as 6–11% in absolute terms for $m = 10,000$ while still maintaining the FDR. The gains over the *Storey (2002)* approach are much more modest in scenarios II–IV, as expected (0–2% in absolute terms while maintaining the FDR for $m = 10,000$). In scenario V, where the covariate is highly informative, the gains in power of our approach over both BH and Storey are much higher. For the Beta(1,20) case, the difference in TPR is threefold for $m = 1,000$ and fivefold for $m = 10,000$ over Storey. Even for the other cases, which may be more realistic, the differences are between 5% and 9% in absolute TPR over Storey and as high as >20% in absolute TPR over BH.

To further explore the potential gain in power over the Storey approach, we expanded scenario V to other functions $\pi_0(x_{i1}) = x_{i1}^k$, where the exponent $k \in \{1, 1.25, 1.5, 2, 3\}$. The $k = 1$ case corresponds to scenario V and used a linear function in the logistic regression, whereas the remaining cases used cubic splines with three degrees of freedom. The estimated FDR and TPR for our approach compared to Storey are shown in Figs. 6 and 7. We note that FDR control is maintained and that in all the simulations, the TPR for our approach is better compared to that for the Storey approach. The gain in

**Table 2 Simulation results for $m = 1,000$ features, 200 runs for each scenario, independent test statistics.**

| | | | FDR % | | | | | TPR % | | | | |
|---|---|---|---|---|---|---|---|---|---|---|---|---|
| $\pi_0(x)$ | Distribution under $H_1$ | Reg. model | BL | Scott T | Scott E | Storey | BH | BL | Scott T | Scott E | Storey | BH |
| I | Beta (1,20) | Linear | 5.0 | | | 5.2 | 3.9 | 0.2 | | | 0.2 | 0.1 |
| II | Beta (1,20) | Linear | 4.8 | | | 4.8 | 4.1 | 0.2 | | | 0.1 | 0.1 |
| II | Beta (1,20) | Spline | 6.5 | | | 4.8 | 4.1 | 0.2 | | | 0.1 | 0.1 |
| III | Beta (1,20) | Linear | 5.2 | | | 5.4 | 5.4 | 0.2 | | | 0.2 | 0.2 |
| III | Beta (1,20) | Spline | 6.2 | | | 5.4 | 5.4 | 0.3 | | | 0.2 | 0.2 |
| IV | Beta (1,20) | Linear | 6.4 | | | 5.1 | 3.4 | 12.2 | | | 5.4 | 0.3 |
| IV | Beta (1,20) | Spline | 7.9 | | | 5.1 | 3.4 | 15.4 | | | 5.4 | 0.3 |
| V | Beta (1,20) | Linear | 3.5 | | | 4.9 | 3.1 | 66.6 | | | 20.6 | 0.4 |
| I | Normal | Linear | 5.0 | 5.2 | 6.6 | 4.9 | 4.4 | 51.0 | 50.9 | 49.7 | 50.8 | 49.7 |
| II | Normal | Linear | 5.4 | 5.7 | 8.1 | 5.3 | 4.9 | 48.5 | 63.5 | 61.3 | 47.6 | 47.0 |
| II | Normal | Spline | 5.6 | 5.9 | 8.3 | 5.3 | 4.9 | 49.3 | 63.5 | 61.5 | 47.6 | 47.0 |
| III | Normal | Linear | 5.8 | 5.9 | 9.9 | 5.4 | 5.1 | 45.1 | 60.3 | 57.9 | 44.0 | 43.4 |
| III | Normal | Spline | 5.9 | 6.0 | 10.1 | 5.4 | 5.1 | 45.6 | 60.9 | 58.2 | 44.0 | 43.4 |
| IV | Normal | Linear | 5.0 | 4.9 | 2.4 | 4.7 | 2.8 | 71.6 | 71.8 | 60.6 | 71.2 | 65.4 |
| IV | Normal | Spline | 5.2 | 5.0 | 2.4 | 4.7 | 2.8 | 72.0 | 71.9 | 60.7 | 71.2 | 65.4 |
| V | Normal | Linear | 4.4 | 4.8 | 21.4 | 4.7 | 2.4 | 79.2 | 83.2 | 73.4 | 74.1 | 67.1 |
| I | T | Linear | 5.7 | 21.3 | 23.4 | 5.5 | 4.8 | 15.7 | 55.4 | 56.9 | 15.2 | 13.6 |
| II | T | Linear | 4.8 | 20.7 | 23.8 | 5.0 | 4.4 | 13.0 | 64.5 | 65.5 | 11.6 | 10.6 |
| II | T | Spline | 4.7 | 21.1 | 24.5 | 5.0 | 4.4 | 13.8 | 64.8 | 65.6 | 11.6 | 10.6 |
| III | T | Linear | 6.2 | 26.8 | 31.0 | 5.9 | 5.4 | 9.4 | 54.6 | 54.7 | 8.2 | 7.6 |
| III | T | Spline | 6.8 | 27.3 | 31.3 | 5.9 | 5.4 | 10.0 | 55.2 | 55.3 | 8.2 | 7.6 |
| IV | T | Linear | 5.0 | 9.3 | 2.8 | 4.7 | 2.9 | 52.5 | 72.9 | 44.4 | 52.0 | 40.3 |
| IV | T | Spline | 5.4 | 9.3 | 2.8 | 4.7 | 2.9 | 53.0 | 73.0 | 44.6 | 52.0 | 40.3 |
| V | T | Linear | 4.1 | 7.4 | 7.8 | 4.7 | 2.5 | 66.4 | 80.3 | 50.0 | 57.1 | 43.3 |
| I | Chisq 1 d$f$ | Linear | 5.0 | | | 4.8 | 4.4 | 51.2 | | | 50.9 | 49.7 |
| II | Chisq 1 d$f$ | Linear | 4.8 | | | 4.8 | 4.4 | 48.3 | | | 47.1 | 46.3 |
| II | Chisq 1 d$f$ | Spline | 5.0 | | | 4.8 | 4.4 | 48.9 | | | 47.1 | 46.3 |
| III | Chisq 1 d$f$ | Linear | 5.0 | | | 4.9 | 4.8 | 44.3 | | | 43.1 | 42.5 |
| III | Chisq 1 d$f$ | Spline | 5.3 | | | 4.9 | 4.8 | 44.8 | | | 43.1 | 42.5 |
| IV | Chisq 1 d$f$ | Linear | 5.1 | | | 4.7 | 2.8 | 71.6 | | | 71.1 | 65.1 |
| IV | Chisq 1 d$f$ | Spline | 5.3 | | | 4.7 | 2.8 | 71.9 | | | 71.1 | 65.1 |
| V | Chisq 1 d$f$ | Linear | 4.4 | | | 4.8 | 2.5 | 78.9 | | | 73.9 | 66.8 |
| I | Chisq 4 d$f$ | Linear | 5.3 | | | 5.4 | 4.8 | 30.8 | | | 30.6 | 29.6 |
| II | Chisq 4 d$f$ | Linear | 5.3 | | | 5.3 | 5.0 | 28.4 | | | 27.5 | 26.7 |
| II | Chisq 4 d$f$ | Spline | 5.4 | | | 5.3 | 5.0 | 29.2 | | | 27.5 | 26.7 |
| III | Chisq 4 d$f$ | Linear | 5.9 | | | 5.4 | 5.3 | 24.8 | | | 24.0 | 23.4 |
| III | Chisq 4 d$f$ | Spline | 5.9 | | | 5.4 | 5.3 | 25.2 | | | 24.0 | 23.4 |
| IV | Chisq 4 d$f$ | Linear | 5.1 | | | 4.7 | 2.8 | 52.3 | | | 51.7 | 44.5 |

| π₀(x) | Distribution under H₁ | Reg. model | FDR % BL | Scott T | Scott E | Storey | BH | TPR % BL | Scott T | Scott E | Storey | BH |
|---|---|---|---|---|---|---|---|---|---|---|---|---|
| IV | Chisq 4 d*f* | Spline | 5.5 | | | 4.7 | 2.8 | 52.7 | | | 51.7 | 44.5 |
| V | Chisq 4 d*f* | Linear | 4.0 | | | 4.6 | 2.4 | 62.8 | | | 55.3 | 46.2 |

**Notes:**
A nominal FDR = 5% was considered. Results for the Scott approaches are only presented for scenarios which generate *z*-statistics or t-statistics.
"Reg. model", specific logistic regression model considered; BL, Boca–Leek; Scott T, Scott theoretical null; Scott E, Scott empirical null; BH, Benjamini–Hochberg.

**Table 3 Simulation results for *m* = 10,000 features, 200 runs for each scenario, independent test statistics.**

| π₀(x) | Distribution under H₁ | Reg. model | FDR % BL | Scott T | Scott E | Storey | BH | TPR % BL | Scott T | Scott E | Storey | BH |
|---|---|---|---|---|---|---|---|---|---|---|---|---|
| I | Beta(1,20) | Linear | 3.7 | | | 3.7 | 3.6 | 0.0 | | | 0.0 | 0.0 |
| II | Beta(1,20) | Linear | 3.1 | | | 3.1 | 3.0 | 0.0 | | | 0.0 | 0.0 |
| II | Beta(1,20) | Spline | 3.1 | | | 3.1 | 3.0 | 0.0 | | | 0.0 | 0.0 |
| III | Beta(1,20) | Linear | 4.0 | | | 3.5 | 3.5 | 0.0 | | | 0.0 | 0.0 |
| III | Beta(1,20) | Spline | 4.5 | | | 3.5 | 3.5 | 0.0 | | | 0.0 | 0.0 |
| IV | Beta(1,20) | Linear | 4.4 | | | 4.8 | 2.5 | 1.2 | | | 0.5 | 0.0 |
| IV | Beta(1,20) | Spline | 5.0 | | | 4.8 | 2.5 | 2.0 | | | 0.5 | 0.0 |
| V | Beta(1,20) | Linear | 3.1 | | | 5.1 | 2.3 | 66.7 | | | 13.1 | 0.0 |
| I | Normal | Linear | 5.0 | 5.0 | 5.9 | 5.0 | 4.5 | 50.6 | 50.6 | 52.1 | 50.7 | 49.6 |
| II | Normal | Linear | 4.9 | 5.2 | 5.3 | 4.9 | 4.6 | 48.4 | 63.9 | 62.9 | 47.3 | 46.6 |
| II | Normal | Spline | 4.9 | 5.2 | 5.3 | 4.9 | 4.6 | 48.8 | 64.0 | 63.0 | 47.3 | 46.6 |
| III | Normal | Linear | 4.9 | 5.2 | 5.5 | 4.9 | 4.7 | 44.2 | 60.2 | 59.3 | 43.5 | 43.0 |
| III | Normal | Spline | 4.9 | 5.2 | 5.4 | 4.9 | 4.7 | 44.4 | 60.6 | 59.7 | 43.5 | 43.0 |
| IV | Normal | Linear | 4.8 | 5.0 | 2.3 | 4.8 | 2.8 | 71.3 | 71.8 | 62.2 | 71.2 | 65.3 |
| IV | Normal | Spline | 4.8 | 5.0 | 2.3 | 4.8 | 2.8 | 71.3 | 71.8 | 62.2 | 71.2 | 65.3 |
| V | Normal | Linear | 4.2 | 5.0 | 23.8 | 4.7 | 2.5 | 79.0 | 83.3 | 74.8 | 74.1 | 66.9 |
| I | T | Linear | 5.2 | 21.7 | 20.8 | 5.1 | 4.7 | 14.1 | 55.3 | 53.2 | 14.1 | 12.6 |
| II | T | Linear | 4.6 | 20.0 | 19.9 | 4.9 | 4.5 | 11.5 | 65.7 | 65.4 | 10.2 | 9.2 |
| II | T | Spline | 4.5 | 20.2 | 20.1 | 4.9 | 4.5 | 12.0 | 65.7 | 65.4 | 10.2 | 9.2 |
| III | T | Linear | 4.9 | 24.7 | 26.8 | 5.2 | 5.2 | 6.8 | 62.5 | 63.7 | 6.0 | 5.5 |
| III | T | Spline | 4.8 | 24.8 | 26.9 | 5.2 | 5.2 | 7.0 | 62.6 | 63.9 | 6.0 | 5.5 |
| IV | T | Linear | 4.8 | 9.3 | 1.2 | 4.8 | 2.9 | 51.8 | 72.8 | 28.5 | 51.6 | 40.2 |
| IV | T | Spline | 4.8 | 9.3 | 1.2 | 4.8 | 2.9 | 51.9 | 72.9 | 28.6 | 51.6 | 40.2 |
| V | T | Linear | 3.9 | 7.4 | 7.3 | 4.6 | 2.5 | 66.0 | 80.7 | 41.1 | 57.1 | 43.4 |
| I | Chisq 1 d*f* | Linear | 5.0 | | | 5.0 | 4.5 | 50.7 | | | 50.6 | 49.6 |
| II | Chisq 1 d*f* | Linear | 4.9 | | | 5.0 | 4.6 | 48.2 | | | 47.2 | 46.4 |
| II | Chisq 1 d*f* | Spline | 4.8 | | | 5.0 | 4.6 | 48.6 | | | 47.2 | 46.4 |
| III | Chisq 1 d*f* | Linear | 5.0 | | | 5.0 | 4.8 | 44.0 | | | 43.2 | 42.7 |

(Continued)

| $\pi_0(x)$ | Distribution under $H_1$ | Reg. model | FDR % | | | | | TPR % | | | | |
|---|---|---|---|---|---|---|---|---|---|---|---|---|
| | | | BL | Scott T | Scott E | Storey | BH | BL | Scott T | Scott E | Storey | BH |
| III | Chisq 1 d$f$ | Spline | 5.0 | | | 5.0 | 4.8 | 44.2 | | | 43.2 | 42.7 |
| IV | Chisq 1 d$f$ | Linear | 4.8 | | | 4.8 | 2.8 | 71.1 | | | 71.0 | 65.2 |
| IV | Chisq 1 d$f$ | Spline | 4.8 | | | 4.8 | 2.8 | 71.2 | | | 71.0 | 65.2 |
| V | Chisq 1 d$f$ | Linear | 4.2 | | | 4.7 | 2.5 | 78.9 | | | 73.9 | 66.9 |
| I | Chisq 4 d$f$ | Linear | 5.0 | | | 5.0 | 4.5 | 29.7 | | | 29.7 | 28.7 |
| II | Chisq 4 d$f$ | Linear | 4.9 | | | 5.0 | 4.7 | 28.0 | | | 27.1 | 26.5 |
| II | Chisq 4 d$f$ | Spline | 4.9 | | | 5.0 | 4.7 | 28.4 | | | 27.1 | 26.5 |
| III | Chisq 4 d$f$ | Linear | 5.2 | | | 5.2 | 5.0 | 24.3 | | | 23.6 | 23.2 |
| III | Chisq 4 d$f$ | Spline | 5.2 | | | 5.2 | 5.0 | 24.4 | | | 23.6 | 23.2 |
| IV | Chisq 4 d$f$ | Linear | 4.7 | | | 4.7 | 2.8 | 51.8 | | | 51.7 | 44.8 |
| IV | Chisq 4 d$f$ | Spline | 4.7 | | | 4.7 | 2.8 | 51.9 | | | 51.7 | 44.8 |
| V | Chisq 4 d$f$ | Linear | 3.9 | | | 4.6 | 2.5 | 62.3 | | | 55.5 | 46.7 |

**Notes:**

A nominal FDR = 5% was considered. Results for the Scott approaches are only presented for scenarios which generate $z$-statistics or $t$-statistics.

"Reg. model", specific logistic regression model considered; BL, Boca–Leek; Scott T, Scott theoretical null; Scott E, Scott empirical null; BH, Benjamini–Hochberg.

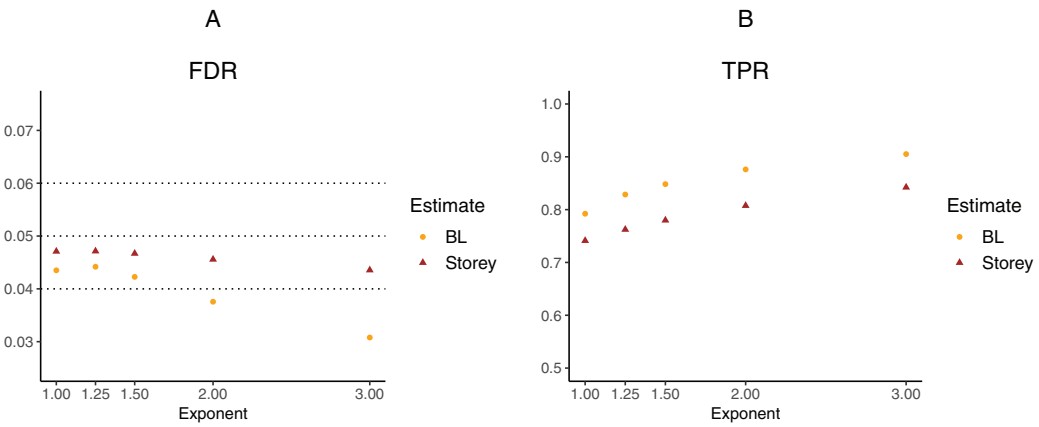

**Figure 6 Simulation results for $m$ = 1,000 features and normally-distributed independent test statistics comparing our proposed approach (BL) to the Storey approach in terms of FDR and TPR.** (A) shows the estimated FDR and (B) the estimated TPR, for a nominal FDR = 5%. Results are averaged over 200 simulation runs. We considered $\pi_0(x_i) = x_i^k$ and varied the exponent $k \in \{1, 1.25, 1.5, 2, 3\}$. 

power is around 5–7% for all the simulations with normally-distributed test statistics (Fig. 6) and around 9–11% for all the simulations with $t$-distributed test statistics (Fig. 7).

Additionally, we explored the case of the "global null", that is, $\pi_0 \equiv 1$. We considered $m$ = 1,000 features, with all the test statistics generated from $N(0,1)$ and 1,000 simulation runs. The mean estimates of $\pi_0(x_{i1})$ are shown in Fig. 8, considering linear terms in both our approach and the Scott T approach. The overall mean for our approach was 0.94, close to the true value of 1 and to the Storey mean estimate of 0.96. At a

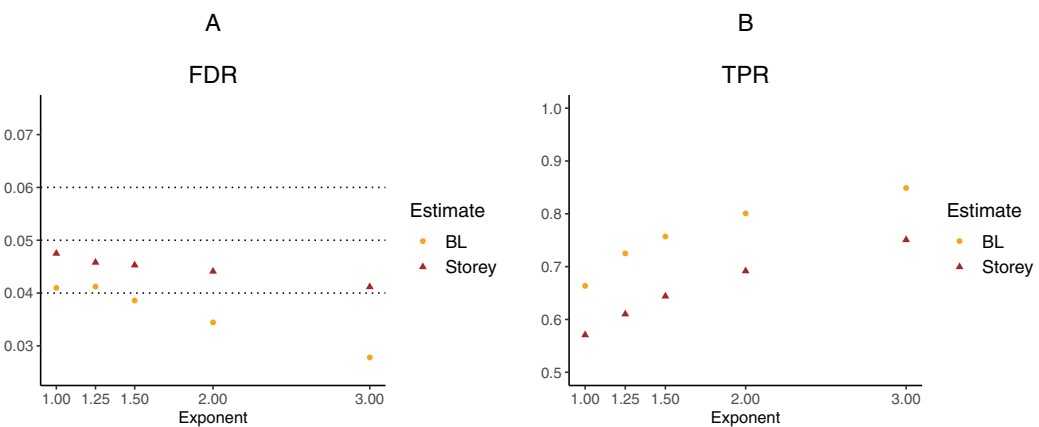

**Figure 7 Simulation results for $m = 1,000$ features and t-distributed independent test statistics comparing our proposed approach (BL) to the Storey approach in terms of FDR and TPR.** (A) shows the estimated FDR and (B) the estimated TPR, for a nominal FDR = 5%. Results are averaged over 200 simulation runs. We considered $\pi_0(x_i) = x_i^k$ and varied the exponent $k \in \{1, 1.25, 1.5, 2, 3\}$.

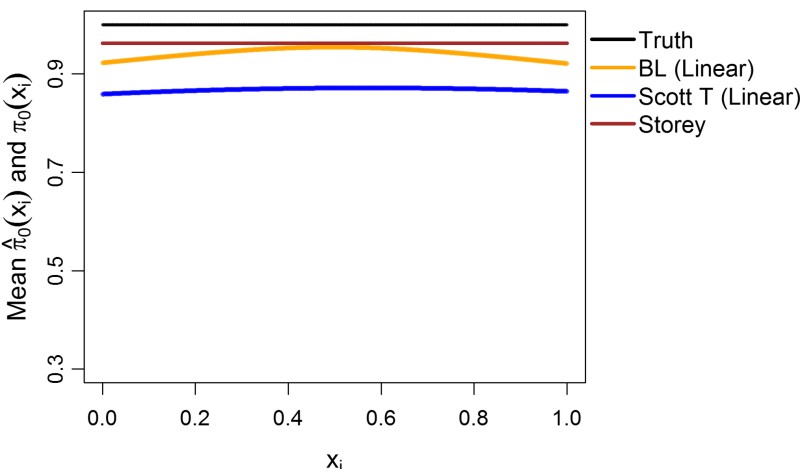

**Figure 8 Simulation results for $m = 1,000$ features, considering the global null $\pi_0 \equiv 1$.** Plot shows the true function $\pi_0(\mathbf{x}_i)$ in black and the empirical means of $\hat{\pi}_0(\mathbf{x}_i)$, assuming different modeling approaches in orange (for our approach, Boca–Leek = BL), blue (for the Scott approach with the theoretical null = Scott T), and brown (for the Storey approach).

nominal FDR of 5%, our approach had an estimated FDR of 5.2%, Scott T of 1.7%, Scott empirical of 21.4%, Storey of 5%, and BH of 4.5%. Interestingly, although the Scott T approach is conservative in terms of the FDR, the estimate of $\pi_0(x_{i1})$ is lower than the estimate obtained from our method, on average. Results were similar when considering splines (5.3% for our approach, 2.1% for Scott T, 22.3% for Scott E).

Finally, we used simulations to explore what happens when there are deviations from independence. Tables S3 and S4 consider simulation results for $m = 1,000$ features and several dependence structures for the test statistics (200 simulation runs per scenario).

We considered multivariate normal and $t$ distributions, with the means drawn as before and block-diagonal variance–covariance matrices with the diagonal entries equal to 1 and a number of blocks equal to either 20 (50 features per block) or 10 (100 features per block). The within-block correlations, $\rho$, were set to 0.2, 0.5, or 0.9. For the multivariate normal distribution, as expected, the FDR was generally closer to the nominal value of 5% for 20 blocks than for 10 blocks, as 20 blocks leads to less correlation. Increasing $\rho$ also leads to worse control of the FDR. Interestingly, for the multivariate $t$ distribution, our method often results in conservative FDRs, with the exception of the spline models and of the case with 10 blocks and $\rho = 0.9$. These same trends are also present for the *Scott et al. (2015)* approaches, but generally with worse control. Furthermore, for $\rho = 0.5$, the empirical null leads to errors in 1% or fewer of the simulation runs; however, for $\rho = 0.9$ it leads to errors in as many as 33% of the runs. In contrast, the *Storey (2002)* method shows estimated FDR values closer to 5% and results in a single error for $\rho = 0.9$ and 10 blocks for the $t$ distribution. We also note that the TPR is generally very low for the multivariate $t$ distributions, except in scenarios IV and V. Overall, while control of the FDR is worse with increasing correlation, as would be anticipated, it is still <0.09 for a nominal value of 0.05 for all scenarios with $\rho \in \{0.2, 0.5\}$, with the control being even better when the estimation uses linear, as opposed to spline, terms.

## REPRODUCIBILITY

All analyses and simulations in this paper are fully reproducible and the code is available on GitHub (https://github.com/SiminaB/Fdr-regression).

## DISCUSSION

We have introduced an approach to estimating FDRs conditional on covariates in a multiple testing framework, by first estimating the proportion of true null hypotheses via a regression model—a method implemented in the `swfdr` package—and then using this in a plug-in estimator. This plug-in approach was also used in *Li & Barber (2017)*, although the estimation procedure therein for $\pi_0(\mathbf{x}_i)$ is different, involving a more complicated constrained maximum likelihood solution; it also requires convexity of the set of possible values of $\pi_0(\mathbf{x}_i)$, which is only detailed in a small number of cases (order structure, group structure, low total variation, or local similarity). One specific caveat is that multiplying by the estimate of $\pi_0(\mathbf{x}_i)$ is equivalent to weighing by $1/\pi_0(\mathbf{x}_i)$, which has been shown to not be Bayes optimal (*Lei & Fithian, 2018*). However, we note that our approach has good empirical properties—further work may consider using our estimate with different weighting schemes.

Our motivating case study considers a GWAS meta-analysis of BMI–SNP associations, where we are interested in adjusting for sample sizes and allele frequencies of the individual SNPs. Using extensive simulations, we compared our approach to FDR regression as proposed by *Scott et al. (2015)*, as well as to the approaches of *Benjamini & Hochberg (1995)* and *Storey (2002)*, which estimate the FDR without covariates. While the *Scott et al. (2015)* approaches outperform our approach for normally-distributed test statistics, which is one of modeling assumptions therein, that approach tends to lose FDR

control for test statistics from the $t$-distribution and cannot be applied in cases where the test statistics come from other distributions, such as the Chi-squared distribution, which may arise from commonly performed analyses; the loss of FDR control for $t$-statistics has been pointed out before for this approach (*Ignatiadis et al., 2016*). In general, our method provides the flexibility of performing the modeling at the level of the $p$-values. Our approach always shows a gain in TPR over the method of *Benjamini & Hochberg (1995)*; the gains over the *Storey (2002)* approach were more modest, but did rise to 5–11% in absolute TPR in cases where the covariates were especially informative. Furthermore, considering a regression context allows for improved modeling flexibility of the proportion of true null hypotheses; future work may build on this method to consider different machine learning approaches in the case of more complicated or high-dimensional covariates of interest. We further show that control of the FDR is maintained in the presence of moderate correlation between the test statistics. We also note that we generally considered models that we thought researchers could be believably interested in fitting—not necessarily the exact models used to generate the simulated data—and our simulations generally showed robustness to misspecifications, including when fitting splines instead of linear terms and in the global null scenario. While beyond the scope of this work, we believe that the issue of model selection will become extremely important as the number of meta-data covariates available increases.

Applying our estimator to GWAS data from the GIANT consortium demonstrated that, as expected, the estimate of the fraction of null hypotheses decreases with both sample size and MAF. It is a well-known and problematic phenomenon that $p$-values for all features decrease as the sample size increases. This is because the null is rarely precisely true for any given feature. One interesting consequence of our estimates is that we can calibrate what fraction of $p$-values appear to be drawn from the non-null distribution as a function of sample size, potentially allowing us to quantify the effect of the "large sample size means small $p$-values" problem directly. Using an FDR cutoff of 5%, our approach leads to 13,384 discoveries, compared to 12,771 from the *Storey (2002)* method; given the fact that they are both multiplicative factors to the *Benjamini & Hochberg (1995)* approach, which in effect assumes the proportion of true null hypotheses to be 1, they both include the 12,500 discoveries using this approach. Thus, our approach leads to additional insights due to incorporating modeling of the fraction of null hypotheses on covariates, as well as to a number of new discoveries. By contrast, the *Scott et al. (2015)* approach leads to very different results based on whether the theoretical null or empirical null is assumed.

We note that our approach relies on a series of assumptions, such as independence of $p$-values and independence of the $p$-values and the covariates conditional on the null. Assuming that the $p$-values are independent of the covariates conditional on the null is also an assumption used by *Ignatiadis et al. (2016)*. Therein, diagnostic approaches for checking this assumption are provided, namely examining the histograms of $p$-values stratified on the covariates. In particular, it is necessary for the distribution to be approximately uniform for larger $p$-values. We perform this diagnostic check in Fig. S3

and note that it appears to hold approximately. The slight conservative behavior seen for smaller values of $N$ in Fig. 1 and Fig. S3 may be the result of publication bias, with SNPs that have borderline significant $p$-values potentially being more likely to be considered in additional studies and thus becoming part of larger meta-analyses. It is interesting that the estimated proportion of nulls in Fig. 2 also starts decreasing substantially right at the median sample size (of 235,717). This may also be due to the same publication bias. Modeling the dependence of $\pi_0$ on meta-data covariates can thus be a good starting place for understanding possible biases and planning future studies.

In conclusion, our approach shows good performance across a range of scenarios and allows for improved interpretability compared to the *Storey (2002)* method. In contrast to the *Scott et al. (2015)* approaches, it is applicable outside of the case of normally distributed test statistics. It always leads to an improvement in estimating the TPR compared to the now-classical *Benjamini & Hochberg (1995)* method, which becomes more substantial when the proportion of null hypotheses is lower. While in very high correlation cases, our method does not appropriately control the FDR, we note that in practice methods are often used to account for such issues at the initial modeling stage, meaning that we generally expect good operating characteristics for our approach. In particular, for GWAS, correlations between sets of SNPs (known as linkage disequilibrium) are generally short-range, being due to genetic recombination during meiosis (*International HapMap Consortium, 2007*); longer-range correlations can result from population structure, which can be accounted for with approaches such as the genomic control correction (*Devlin & Roeder, 1999*) or principal components analysis (*Price et al., 2006*). For gene expression studies, batch effects often account for between-gene correlations; many methods exist for removing these (*Johnson, Li & Rabinovic, 2007*; *Leek & Storey, 2007*; *Leek, 2014*). We also note the subtle issue that the accuracy of the estimation is based on the number of features/tests considered, not on the sample sizes within the tests. Thus, our "large-sample" theoretical results are to be interpreted within this framework. In our simulations, for example, we see that using 10,000 rather than 1,000 features improved the FDR control. In particular, the models with splines estimated a larger number of parameters, leading to poorer FDR control for the case with a smaller number of features; there is also worse control for spline models when simulating dependent statistics, as the effective number of features in that case is even smaller. Thus, in general we recommend considering simpler models in scenarios that have a small number of features. We note that our motivating data analysis had over 2.5 million features and that many high-dimensional problems have features in the tens of thousands or higher. A range of other applications for our methodology are also possible by adapting our regression framework, including estimating FDRs for gene sets (*Boca et al., 2013*), estimating science-wise FDRs (*Jager & Leek, 2013*), or improving power in high-throughput biological studies (*Ignatiadis et al., 2016*). Thus, this is a general problem and as more applications accumulate, we anticipate our approach being increasingly used to provide additional discoveries and scientific insights.

### Funding

This work was supported by a grant from NIH R01 GM105705. The funders had no role in study design, data collection and analysis, decision to publish, or preparation of the manuscript.

### Grant Disclosure

The following grant information was disclosed by the authors:
Grant from NIH: R01 GM105705.

### Competing Interests

The authors declare that they have no competing interests.

### Author Contributions

- Simina M. Boca conceived and designed the experiments, performed the experiments, analyzed the data, contributed reagents/materials/analysis tools, prepared figures and/or tables, authored or reviewed drafts of the paper, approved the final draft.
- Jeffrey T. Leek conceived and designed the experiments, contributed reagents/materials/ analysis tools, authored or reviewed drafts of the paper, approved the final draft.

### Data Availability

GitHub: https://github.com/SiminaB/Fdr-regression.

### Supplemental Information

Supplemental information for this article can be found online at http://dx.doi.org/10.7717/peerj.6035#supplemental-information.

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
