# Peer review of "A direct approach to estimating false discovery rates conditional on covariates"

_PeerJ, doi:10.7717/peerj.6035_

## Round 0.1 · original submission · Major Revisions

Both reviewers agree that the work is well-motivated, technically sound and practically useful. They also offered many suggestions to improve the method/manuscript including de-emphasizing of the proposed modified BH procedure, comparing to some state-of-the-art of method (IHW), conducting more robustness studies, providing guidance against potential FDR inflation and citing relevant literature. Please address these comments by additional simulations or further explanations.

Reviewer 1 ·

Basic reporting

Please see the report attached.

Experimental design

Please see the report attached.

Validity of the findings

Please see the report attached.

Additional comments

Please see the report attached.

Annotated reviews are not available for download in order to protect the identity of reviewers who chose to remain anonymous.

·

Basic reporting

no comment

Experimental design

no comment

Validity of the findings

no comment

Additional comments

The authors present a framework for estimating the false discovery rate conditionally on covariates, when the p-values are independent of these under the null hypothesis. In the analysis of large scale datasets, this is an important, challenging and potentially valuable task. The paper is well-written, well-motivated, and technically sound.

The underlying theme of this paper could be split into two parts: First, the authors propose an estimator of the proportion of null hypotheses conditionally on covariates. Second, they use the estimator to modify the standard Benjamini-Hochberg procedure. We remark on both aspects:

A. Estimation of $\pi_0(X_i)$:
* * *
The first part, i.e., estimation of $\pi_0(X_i)$ is, in our opinion, the selling point of this paper. The authors phrase the task of $\pi_0(X_i)$ estimation as a regression task. This key observation opens up many opportunities; in this paper the authors explore the use of linear and logistic regression potentially in conjunction with regression splines. However, their idea also enables the use of machine learning methods (say random forests, boosting, neural networks) to estimate $\pi_0(X_i)$, for example, if the covariates are complicated (say sequences or images) and high-dimensional. This is not mentioned explicitly in the paper, and we believe it would be worth mentioning.

Furthermore, the authors emphasize the importance of estimating $\pi_0(X_i)$ for tasks not directly related to down-stream multiple testing. For example, in the discussion they attribute the conservative behaviour in the right panel of Fig. 1 to publication bias. It would be interesting if this could be discussed in more detail.

Some missing references for this section include: the SABHA procedure [1], which also uses the thresholded p-values to estimate $\pi_0(X_i)$ but employs a maximum likelihood approach + convex optimization instead; AdaPT [2], which uses EM and logistic regression to estimate $\pi_0(X_i)$; and for categorical covariates, as in supplement S2.2., the GBH [3] procedure, which uses a standard $\pi_0$ estimator within each level.

It could also be interesting to see the following: How does averaging all the $\hat{\pi}_0(X_i)$ perform compared to $\hat{\pi_0}$, if we are just interested in estimating the global $\pi_0$?

B. Downstream multiple testing with FDR control:
* * *
Another large part of this paper is concerned with modifying the BH procedure to account for $\pi_0(X_i)$ estimation. This is achieved by multiplying the adjusted p-values by the corresponding conditional $\pi_0$ estimate. We feel that this part of the paper is not as well motivated and could potentially be drastically deemphasized or even removed, similarly to an older version of this manuscript [4].

The main caveats of this section are as follows: The proposed scheme of modifying BH is essentially the same as that proposed in the SABHA procedure [1]. Second, as mentioned in [2] and [5], this scheme is inadmissible even under oracle knowledge of $\pi_0(X_i)$. Instead one should use a weighted BH scheme. One choice would be to use weighted BH with weights equal to $(1-\pi_0(X_i))/\pi_0(X_i)/(1-\pi_{0,\text{global}})$. This is the scheme proposed for categorical covariates in the GBH [3] procedure, and it leads to power increase both by upweighting the right hypotheses and by being adaptive (alpha-exhaustive). It would be important to see how such a scheme performs compared to the method presented here; we think it would further illustrate why task A. is so important. A further option would be to also use weights proportional to $(1-\pi_0(X_i))/\pi_0(X_i)$, but normalized to average to 1. Then the procedure would no longer be adaptive, but it could also be used with the cross-weighting scheme in [5] and would automatically enjoy finite sample FDR control guarantees.



Minor remarks:
* * *
* After Eqn. (1): "Namely the expected fraction": Conditional on at least making one rejections?

* $\theta_i = 1$ when null is a bit confusing and non-standard notation, maybe change it to $\theta_i = 0$.

* Page 5 typo "In section S3 we provide theoretical results for this estimator in Section S2 of the suppl. materials"

* Why not try the model with the cubic spline for Scenario 1 as well?

* It would be nice to see simulations under the global null as well.

* In the discussion it is mentioned that the method of Scott (2015) often does not control FDR. This was also mentioned/shown in [6]

* Proof of Supplement S3: Some parentheses missing after "then we can combine them as follows"


This review was prepared by Nikos Ignatiadis together with Wolfgang Huber.


References:
* * *
[1] Li, Ang, and Rina Foygel Barber. "Multiple testing with the structure adaptive Benjamini-Hochberg algorithm." arXiv preprint arXiv:1606.07926 (2016).

[2] Lei, Lihua, and William Fithian. "AdaPT: an interactive procedure for multiple testing with side information." Journal of the Royal Statistical Society: Series B (Statistical Methodology) (2018).

[3] Hu, James X., Hongyu Zhao, and Harrison H. Zhou. "False discovery rate control with groups." Journal of the American Statistical Association 105.491 (2010): 1215-1227.

[4] Boca, Simina M., and Jeffrey T. Leek. "A regression framework for the proportion of true null hypotheses." bioRxiv (2015): 035675.

[5] Ignatiadis, Nikolaos, and Wolfgang Huber. "Covariate-powered weighted multiple testing." arXiv preprint arXiv:1701.05179 (2018).

[6] Ignatiadis, Nikolaos, et al. "Data-driven hypothesis weighting increases detection power in genome-scale multiple testing." Nature Methods 13.7 (2016): 577.

---

## Round 0.2 · accepted · Accept

The reviewer is satisfied with the current revision.

One minor comment: I noticed that you wrote that "The [22] methods use z-values, as opposed to the other methods, which use p-values; as a result, in this case we input the t-statistics into the Scott T approach, leading to a more pronounced anti-conservative behavior in some cases. " It seems more reasonable to transform the p-value into z-value (i.e., \phi^{-1}(1 - pvalue)) before applying FDRreg.

# Reviewer 1 ·

Basic reporting

No comment.

Experimental design

No comment.

Validity of the findings

No comment.

Additional comments

I am satisfied with the revision and would recommend it for publication.

---

## Author Rebuttal · Round 0.2

# Response to reviewers for "A direct approach to estimating false discovery rates conditional on covariates"

Simina M. Boca and Jeffrey T. Leek

October 10, 2018

We thank the reviewers and the editor for their comments. We are particularly pleased that all reviewers find the paper "conceptually sound" with "the potential to make an impact in real applications" (Reviewer 1), respectively "well-written, well-motivated, and technically sound" (Reviewers 2). Our main changes to this revised version have been as follows:

- We included the additional literature that the reviewers pointed out - we agree that this was an oversight on our part initially. This is an exciting and increasingly crowded field, with several papers between 2015 and 2018. In fact, the first version of our preprint, which only performed the estimation of $\pi_0(x)$, was posted in 2015.

- We included additional scenarios where our method shows a more substantial gain in power over the Storey approach, as requested by Reviewer 1.

- We emphasized the estimation of $\pi_0(x)$ more, as desired by Reviewers 2.

We recognize that there are an increasing number of methods available to perform this type of estimation. As a "Bioinformatics tool" paper, we find our accompanying package to be a particularly valuable contribution to this field. We are also aware of at least one effort that is underway to perform thorough comparisons of all these possible approaches.

Below, the reviewers' words are marked in *italics*. New additions or changes to the manuscript are marked in red.

## Response to reviewer 1

*This paper proposes a regression approach to estimate the varying proportion of null hypotheses conditional on external covariates. The estimated null proportions are then used to adjust the cutoff values in the BH procedure to incorporate external covariate information. The method can be viewed as a special case of the weighted BH procedure (Genovese et*

*al. 2006) with the weights estimated using logistic regression. Although the method is conceptually sound and has the potential to make an impact in real applications, there are a few issues that need to be addressed carefully before the paper can be considered for publication. Below are some specific comments and suggestions on the manuscript.*

1. *There are some recent works on similar topics (e.g., Li and Barber 2017; Lei and Fithian 2017) which the authors did not cite.*

   - Thank you. We have now added these references. In the Introduction section, we now state:

     "Similar very recent approaches include work by Li and Barber (2017) and Lei and Fithian (2018), which also estimate $\pi_0$ based on existing covariates, using different approaches. The approach of Ignatiadis et al (2016) considers p-value weighting but conservatively estimates $\pi_0 \equiv 1$. An overview of the differences between these various approaches for incorporating meta-data and the relationships between them is provided in Ignatiadis and Huber (2018)."

   We have also added the reference to the Genovese et al paper:

   "The concept of using these feature-level covariates, which may also be considered "prior information," arose in the context of p-value weighting (Genovese at al., 2006)."

2. *The numerical results are not very encouraging. More convincing results are needed to demonstrate the usefulness of the new method over John Storey's approach when the covariates are informative.*

   - We have now added a number of scenarios where we show a 5-11% absolute improvement in terms of power (TPR) over Storey's approach. Specifically, we added Scenario V, where the function is $\pi_0(x_i) = x_i$, to all the simulations. The results for Scenario V are presented in Tables 2-3 and S3-S4. We also considered $\pi_0(x_i) = x_i^k$, for $k \in \{1, 1.25, 1.5, 2, 3\}$ for the scenarios with m=1,000 features and either normally-distributed or t-distributed statistics and performed comparisons between our approach and Storey's approach, illustrated in Figures 6-7. We now state in the Simulations section:

     "In scenario V, where the covariate is highly informative, the gains in power of our approach over both BH and Storey are much higher. For the Beta(1,20) case, the difference in TPR is threefold for $m = 1,000$ and fivefold for $m = 10,000$ over Storey. Even for the other cases, which may be more realistic, the differences are between 5% and 9% in absolute TPR over Storey and as high as >20% in absolute TPR over BH.

     To further explore the potential gain in power over the Storey approach, we expanded scenario V to other functions $\pi_0(x_{i1}) = x_{i1}^k$, where the exponent $k \in \{1, 1.25, 1.5, 2, 3\}$. The $k = 1$ case corresponds to scenario

V and used a linear function in the logistic regression, whereas the remaining cases used cubic splines with 3 degrees of freedom. The estimated FDR and TPR for our approach compared to Storey are shown in Figures 6 and 7. We note that FDR control is maintained and that in all the simulations, the TPR for our approach is better compared to that for the Storey approach. The gain in power is around 5-7% for all the simulations with normally-distributed test statistics (Figure 6) and around 9-11% for all the simulations with t-distributed test statistics (Figure 7)."

3. *The authors may want to compare their method with the independent hypothesis weighting procedure proposed in Ignatiadis et al. (2016) in simulation studies.*

   - Thank you for this suggestion. We feel like this is beyond the scope of this current work. In particular, we note that we are estimating $\pi_0$, whereas Ignatiadis et al. (2016) does not estimate it and conservatively assumes that it is 1. Ignatiadis and Huber (2018, arXiv) actually make this point. Note our added text in the introduction, referenced in point 1 above.

4. *The assumption of Theorem S1 should be strengthened. It seems the authors assumed a mixture model for $p_i$ conditional on $x_i : pi(x_i)Unif(0,1) + (1 - pi(x_i))G_{x_i}$. This is stronger than what is assumed in Theorem S1 (i.e., conditional on the null, the p-values do not depend on the covariates).*

   - Thank you for pointing this out. We had noted this right above Theorem S1, but it should have been part of the theorem assumptions. The statement of Theorem S1 now includes:

     "Furthermore, the null p-values have a Uniform(0,1) distribution, whereas the alternative p-values have a distribution with cdf $G_{\mathbf{x}_i}$, as defined above."

     We also made the following change to Section 4, which describes the algorithm:

     "Our theoretical results are based on the more restrictive assumption that the null p-values have a Uniform$(0, 1)$ distribution, whereas the distribution of the alternative p-values may depend on the covariates."

5. *The idea discussed in Section S2.2 has already been proposed in the statistics literature, see, e.g., Hu et al. (2010).*

   - Thank you. We have now added the following sentence in Section S2.2:

     "This idea has been proposed before, for example in Hu et al. (2010), but we propose it here as a natural subcase of our approach."

     We also changed the reference in the main manuscript to Hu et al. (2010):

"in Section S2, we detail how the case of no covariates and the case where the features are partitioned into sets, such as in Hu et al. (2010), can be seen as special cases of our more general framework when the linear regression approach is applied;"

6. *A weakness of the method is that it is only valid (in theory) for correctly specified models. From a practical viewpoint, it is of great interest to study the robustness of the procedure for misspecified models.*

- In our current work, we are studying the robustness via simulations. For the simulations, we are always considering relatively standard models, namely either linear terms or splines. In particular, for scenarios II-IV, we are never using the perfectly correctly specified model (this would regardless be impossible given that we are using logistic regression to perform the estimation), but rather models that we think individual researchers would be likely to fit. We are also considering robustness to the independence assumption. We have also now added Table S2, that presents the spline results for scenarios I and V, as well as a scenario that considers the global null (plot of estimated means in Figure 8, results discussed on page 8.) These additions were to respond to points made by the other reviewers, but are also important when considering misspecification. We believe that model selection could be a major future direction of research in this field, but is beyond the scope of this work. We have now added to the Discussion:

  "We also note that we generally considered models that we thought researchers could be believably interested in fitting - not necessarily the exact models used to generate the simulated data - and our simulations generally showed robustness to misspecifications, including when fitting splines instead of linear terms and in the global null scenario. While beyond the scope of this work, we believe that the issue of model selection will become extremely important as the number of meta-data covariates available increases."

7. *The spline transformation seems to lead to significant FDR inflation especially when the statistics are dependent. Some guidance on the use of spline transformation on the covariates would be very helpful.*

- This is true, although we do note that it does not get especially bad until we get to very high correlations. We have now emphasized this at the end of the Simulations section:

  "Overall, while control of the FDR is worse with increasing correlation, as would be anticipated, it is still $< 0.09$ for a nominal value of $0.05$ for all scenarios with $\rho \in \{0.2, 0.5\}$, with the control being even better when the estimation uses the linear model."

  We have also added to the Discussion section:

"In particular, the models with splines estimated a larger number of parameters, leading to poorer FDR control for the case with a smaller number of features; there is also worse control for spline models when simulating dependent statistics, as the effective number of features in that case is even smaller. Thus, in general we recommend considering simpler models in scenarios that have a small number of features."

## Response to reviewer 2

*The authors present a framework for estimating the false discovery rate conditionally on covariates, when the p-values are independent of these under the null hypothesis. In the analysis of large scale datasets, this is an important, challenging and potentially valuable task. The paper is well-written, well-motivated, and technically sound.*

*The underlying theme of this paper could be split into two parts: First, the authors propose an estimator of the proportion of null hypotheses conditionally on covariates. Second, they use the estimator to modify the standard Benjamini-Hochberg procedure. We remark on both aspects:*

### A. Estimation of $\pi_0(X_i)$:

*The first part, i.e., estimation of $\pi_0(X_i)$ is, in our opinion, the selling point of this paper. The authors phrase the task of $\pi_0(X_i)$ estimation as a regression task. This key observation opens up many opportunities; in this paper the authors explore the use of linear and logistic regression potentially in conjunction with regression splines. However, their idea also enables the use of machine learning methods (say random forests, boosting, neural networks) to estimate $\pi_0(X_i)$, for example, if the covariates are complicated (say sequences or images) and high-dimensional. This is not mentioned explicitly in the paper, and we believe it would be worth mentioning.*

- Thank you for your input. As we also note below, we have reemphasized the estimation of $\pi_0$ as a function of meta-data covariates. In particular, Figures 4 and 5, which show the estimated means of $\pi_0(\mathbf{x}_i)$ have been moved back to the main manuscript from the supplement and Figure 8, which shows the same in the global null scenario has been added as per one of your later recommendations. We have also added the following to the Discussion:

  "Furthermore, considering a regression context allows for improved modeling flexibility of the proportion of true null hypotheses; future work may build on this method to consider different machine learning approaches in the case of more complicated or high-dimensional covariates of interest."

*Furthermore, the authors emphasize the importance of estimating $\pi_0(X_i)$ for tasks not directly related to downstream multiple testing. For example, in the discussion they attribute the conservative behaviour in the right panel of Fig.*

*1 to publication bias. It would be interesting if this could be discussed in more detail.*

- We modified the text in the Discussion to clarify our point:

  "The slight conservative behavior seen for smaller values of N in Figures 1 and S3 may be the result of publication bias, with SNPs that have borderline significant p-values potentially being more likely to be considered in additional studies and thus becoming part of larger meta-analyses."

  We also added:

  "Modeling the dependence of $\pi_0$ on meta-data covariates can thus be a good starting place for understanding possible biases and planning future studies."

*Some missing references for this section include: the SABHA procedure [1], which also uses the thresholded p-values to estimate $\pi_0(X_i)$ but employs a maximum likelihood approach + convex optimization instead; AdaPT [2], which uses EM and logistic regression to estimate $\pi_0(X_i)$; and for categorical covariates, as in supplement S2.2., the GBH [3] procedure, which uses a standard $\pi_0$ estimator within each level.*

- Thank you. We now include these references. In the Introduction section, we now have:

  "Similar very recent approaches include work by Li and Barber (2017) and Lei and Fithian (2018), which also estimate $\pi_0$ based on existing covariates, using different approaches. The approach of Ignatiadis et al (2016) considers p-value weighting but conservatively estimates $\pi_0 \equiv 1$. An overview of the differences between these various approaches for incorporating meta-data and the relationships between them is provided in Ignatiadis and Huber (2018)."

  In Section S2.2, we now have:

  "This idea has been proposed before, for example in Hu et al. (2010), but we propose it here as a natural subcase of our approach."

  We also changed the reference in the main manuscript to Hu et al. (2010):

  "in Section S2, we detail how the case of no covariates and the case where the features are partitioned into sets, such as in Hu et al. (2010), can be seen as special cases of our more general framework when the linear regression approach is applied;"

*It could also be interesting to see the following: How does averaging all the $\hat{\pi}_0(X_i)$ perform compared to $\hat{\pi}_0$, if we are just interested in estimating the global $\pi_0$?*

- One can see from the results for Scenario I ($\pi_0 \equiv 0.9$) and for the global null ($\pi_0 \equiv 1$), which both represent flat cases, that our estimates, and therefore their averages, are close to the estimates from the Storey method. We also added the following to the Simulations section:

  "For all these simulation frameworks, we note that for scenario I, the overall mean across all simulations for our method was between 0.88 and 0.91, very close to the true value of 0.9."

  "The overall mean for our approach was 0.94, close to the true value of 1 and to the Storey mean estimate of 0.96."

### *B. Downstream multiple testing with FDR control:*

*Another large part of this paper is concerned with modifying the BH procedure to account for $\pi_0(X_i)$ estimation. This is achieved by multiplying the adjusted p-values by the corresponding conditional $\pi_0$ estimate. We feel that this part of the paper is not as well motivated and could potentially be drastically deemphasized or even removed, similarly to an older version of this manuscript [4].*

- Thank you for this input. We have now reemphasized some of the results for the conditional estimate. However, we believe that it is helpful to also include the FDR control results, hence our second part using the plug-in procedure.

*The main caveats of this section are as follows: The proposed scheme of modifying BH is essentially the same as that proposed in the SABHA procedure [1]. Second, as mentioned in [2] and [5], this scheme is inadmissible even under oracle knowledge of $\pi_0(X_i)$. Instead one should use a weighted BH scheme. One choice would be to use weighted BH with weights equal to $(1 - \pi_0(X_i))/\pi_0(X_i)/(1 - \pi_{0,global})$. This is the scheme proposed for categorical covariates in the GBH [3] procedure, and it leads to power increase both by upweighting the right hypotheses and by being adaptive (alpha-exhaustive). It would be important to see how such a scheme performs compared to the method presented here; we think it would further illustrate why task A. is so important. A further option would be to also use weights proportional to $(1 - \pi_0(X_i))/\pi_0(X_i)$, but normalized to average to 1. Then the procedure would no longer be adaptive, but it could also be used with the cross-weighting scheme in [5] and would automatically enjoy finite sample FDR control guarantees.*

- Thank you for this note. We have now noted in the Discussion:

  "This plug-in approach was also used in Li and Barber (2017), although the estimation procedure therein for $\pi_0(\mathbf{x}_i)$ is different, involving a more complicated constrained maximum likelihood solution; it also requires convexity of the set of possible values of $\pi_0(\mathbf{x}_i)$, which is only detailed in a small number of cases (order structure, group structure, low total variation or local similarity). One specific caveat is that multiplying by the estimate of $\pi_0(\mathbf{x}_i)$ is equivalent to weighing by $1/\pi_0(\mathbf{x}_i)$, which has been shown to not be Bayes optimal (Lei and Fithian, 2018).

However, we note that our approach has good empirical properties - further work may consider using our estimate with different weighting schemes."

Thus, we find it beyond the scope of the present work to consider different weighting schemes. Thank you for the suggestions however!

*Minor remarks:*

- *After Eqn. (1): "Namely the expected fraction": Conditional on at least making one rejections?*

    – Yes, you are correct. We have now fixed this.

- $\theta_i = 1$ *when null is a bit confusing and non-standard notation, maybe change it to* $\theta_i = 0$.

    – We made this change.

- *Page 5 typo "In section S3 we provide theoretical results for this estimator in Section S2 of the suppl. materials"*

    – Thank you! We have corrected this.

- *Why not try the model with the cubic spline for Scenario 1 as well?*

    – We have now done this for the case of m=1,000 independent features. We present the results in the new Table S2 in the Supplement.

- *It would be nice to see simulations under the global null as well.*

    – We added this simulation scenario for m=1,000 features. The estimated mean of values are presented in Figure 8. We also added to the Simulations section:

    "Additionally, we explored the case of the "global null," i.e. $\pi_0 \equiv 1$. We considered $m = 1,000$ features, with all the test statistics generated from $N(0, 1)$ and 1,000 simulation runs. The mean estimates of $\pi_0(x_{i1})$ are shown in Figure 8, considering linear models for both our approach and the Scott T approach. At a nominal FDR of 5%, our approach had an estimated FDR of 5.2%, Scott T of 1.7%, Scott empirical of 21.4%, Storey of 5%, and BH of 4.5%. Interestingly, although the Scott T approach is conservative in terms of the FDR, the estimate of $\pi_0(x_{i1})$ is lower than the estimate obtained from our method, on average. Results were similar when considering splines (5.3% for our approach, 2.1% for Scott T, 22.3% for Scott empirical.) "

- *In the discussion it is mentioned that the method of Scott (2015) often does not control FDR. This was also mentioned/shown in [6]*

    - We have now added this to the Discussion: "the loss of FDR control for t-statistics has been pointed out before for this approach (Ignatiadis et al., 2016)."

- *Proof of Supplement S3: Some parentheses missing after "then we can combine them as follows"*

    - The parentheses look OK to us. Do you have a specific concern? We used the "[{()}]" convention.